# Increased Expression of Anaphylatoxin C5a-Receptor-1 in Neutrophils and Natural Killer Cells of Preterm Infants

**DOI:** 10.3390/ijms241210321

**Published:** 2023-06-19

**Authors:** Hannah Boeckel, Christian M. Karsten, Wolfgang Göpel, Egbert Herting, Jan Rupp, Christoph Härtel, Annika Hartz

**Affiliations:** 1Department of Pediatrics, University of Lübeck, 23538 Lübeck, Germanyegbert.herting@uksh.de (E.H.);; 2International Research Training Group 1911, University of Lübeck, 23538 Lübeck, Germanyjan.rupp@uksh.de (J.R.); 3Institute for Systemic Inflammation Medicine, University of Lübeck, 23538 Lübeck, Germany; 4German Center of Infection Research, Hamburg-Lübeck-Borstel-Riems, 23538 Lübeck, Germany; 5Department of Infectious Diseases and Microbiology, University of Lübeck, 23538 Lübeck, Germany; 6Interdisciplinary Center of Clinical Research, University of Würzburg, 97080 Würzburg, Germany; 7Department of Pediatrics, University of Würzburg, 97080 Würzburg, Germany

**Keywords:** preterm infants, C5a, C5aR1, neutrophils, NK cells, innate immunity, sepsis

## Abstract

Preterm infants are susceptible to infection and their defense against pathogens relies largely on innate immunity. The role of the complement system for the immunological vulnerability of preterm infants is less understood. Anaphylatoxin C5a and its receptors C5aR1 and -2 are known to be involved in sepsis pathogenesis, with C5aR1 mainly exerting pro-inflammatory effects. Our explorative study aimed to determine age-dependent changes in the expression of C5aR1 and C5aR2 in neonatal immune cell subsets. Via flow cytometry, we analyzed the expression pattern of C5a receptors on immune cells isolated from peripheral blood of preterm infants (n = 32) compared to those of their mothers (n = 25). Term infants and healthy adults served as controls. Preterm infants had a higher intracellular expression of C5aR1 on neutrophils than control individuals. We also found a higher expression of C5aR1 on NK cells, particularly on the cytotoxic CD56^dim^ subset and the CD56^-^ subset. Immune phenotyping of other leukocyte subpopulations revealed no gestational-age-related differences for the expression of and C5aR2. Elevated expression of C5aR1 on neutrophils and NK cells in preterm infants may contribute to the phenomenon of “immunoparalysis” caused by complement activation or to sustained hyper-inflammatory states. Further functional analyses are needed to elucidate the underlying mechanisms.

## 1. Introduction

Preterm infants have a functionally distinct immune system, which is adapted to the immunotolerogenic situation in utero rather than to a defense against pathogens. This leaves preterm infants more susceptible to infections [1,2], which are major causes of morbidity and mortality [3]. As their adaptive immune responses are regarded as inexperienced (naïve for microbial antigens), defense mechanisms are focused on innate immunity and temporary “tolerogenic” mechanisms allowing a physiological microbiome establishment [4,5].

Neutrophils, important as a “first line of defense” against pathogens by recognizing and degrading bacteria and producing anti-microbial peptides (AMPs), are known to be functionally less efficient in preterm infants [6]. They have reduced phagocytic capabilities [7,8], show an impaired formation of neutrophil extracellular traps (NETs) [9], chemotaxis and bacterial killing, and a reduced capacity to produce reactive oxygen species (ROS) and AMPs [10,11,12,13].

NK cells, also belonging to the innate immune system and important for spontaneous killing of pathogens and cytokine responses, are also hyporesponsive in neonates [9,14]. They have lower cytolytic capacity and antibody-dependent cell cytotoxicity (ADCC) than adult NK cells [9,14], which is a maturational issue [15]. Furthermore, the proportions of NK cell subsets differ in neonates and adults. In preterm infants, the immunoregulatory CD56^bright^ NK cells are decreased compared to full-term infants [16,17]. In addition, neonates have more CD56^−^CD16^+^ NK cells than adults [17], a subset with impaired lytic capacity, which is only detectable in cohorts with reduced immune competence [17,18].

Complement, particularly anaphylatoxin C5a, is a key factor in innate immunity. In sepsis, uncontrolled activation of the complement system [19] leads to substantial generation of C5a [20], which is associated with disease severity, multiorgan failure [21,22] and increased mortality.

Through its two receptors, C5aR1 and C5aR2, C5a exerts a variety of effects. The G-protein-coupled C5aR1 is known to exhibit proinflammatory functions [23,24]. The role of C5aR2 (previously known as GPR77 or C5L2) remains controversial [25,26,27], since pro- as well as anti-inflammatory properties have been demonstrated [28]. C5aR2 was initially suggested to act as a decoy receptor for C5a [29] and modulate C5aR1 signaling [26], prohibiting overactivation of C5aR1. Research groups also suggested a proinflammatory role of the receptor through HMGB1 during sepsis [30]. However, human data are sparse, and so far, no studies exist regarding the role of C5aR1 and C5aR2 in neonatal immunity.

Levels of circulating complement components in neonates are lower than in adults, while data on functionality are scarce. C3- and C4-levels are reduced in preterm infants and correlate inversely with gestational age, while the lytic capacity of the complement system is diminished [31].

In this descriptive study, we investigated the expression of the corresponding receptors C5aR1 and C5aR2 on leucocytes of preterm infants as compared to those of their mothers, term neonates and healthy adult donors.

## 2. Results

### 2.1. Cohort Characteristics

We performed a convenience sample study with blood samples obtained from n = 32 preterm infants (between >23 + 5 and <33 ± 0 weeks of gestation) and their mothers (n = 25). Healthy term infants (n = 11) and healthy adults (n = 14) served as controls.

Clinical characteristics of the study cohort are described in Table 1.

### 2.2. Flow Cytometry

#### 2.2.1. Elevated Intracellular Expression of C5aR1 in Preterm Neutrophils

To distinguish between intracellular and extracellular expression, we split the blood samples of 50 µL each and measured surface expression in one aliquot, while the other aliquot was fixed and permeabilized for measurement of intracellular expression. The percentage of neutrophils expressing C5aR1 intracellularly (Figure 1a) and their MFI of intracellular C5aR1 (Figure 1b) were significantly increased in preterm infants compared to their mothers, adults and term born infants.

In total, 84.10% of preterm neutrophils expressed C5aR1 intracellularly, whereas expression was lower in adults at 15.35% (*p* < 0.0001), mothers at 8.10% (*p* < 0.0001) and term newborns at 34.80% (*p* = 0.004).

The MFI for C5aR1 was also significantly elevated in preterm neutrophils compared to adults (*p* = 0.0009) and mothers (*p* < 0.0001) (Figure 1b).

Regarding the extracellular expression of C5aR1, adult neutrophils expressed significantly more C5aR1 on their surface as compared to preterm infant neutrophils (*p* = 0.027).

C5aR1 expression patterns of neutrophils from preterm infants with early-onset sepsis (EOS) or mothers with amniotic infection syndrome (AIS) did not differ significantly from preterm infants without sepsis and mothers without AIS. We also found no association between C5aR1 expression and gestational age, birth weight or sex.

#### 2.2.2. Density of C5aR1 Expression in NK Cells of Preterm Infants and on Their Surface Is Elevated

As described in Figure 2, NK cells of preterm infants had a higher mean fluorescence intensity of extracellular and intracellular C5aR1 as compared to term infants, mothers, and adult controls (Figure 2).

To determine whether this finding is reflected by a specific subtype of NK cells, gating for different NK cell subsets was applied. This showed an upregulated expression of C5aR1 in the CD56^dim^ and CD56^-^ subsets (Figure 3).

In contrast, CD56^bright^ CD16^+^ NK cells from term and preterm infants showed a significantly lower MFI for C5aR1 compared to mothers and adults (Kruskal–Wallis *p* < 0.0001).

The extracellular expression of C5aR1 on NK cells from preterm infants without infection (median-MFI = 3263) was significantly higher compared to infants with EOS or born to mothers with AIS (median-MFI = 2299) (Mann–Whitney test *p* = 0.009, Figure 4), especially on the CD56^dim^ CD16^-^ subset.

There was no correlation in the expression pattern of C5aR1 on NK cells with gestational age, birth weight or sex.

We also found higher intra- and extracellular mean fluorescence index values for C5aR1 expression in preterm infant B cells and a higher percentage of C5aR1 expression in preterm infants B cells as compared to term infants and adult samples (Appendix A).

There was no correlation of the C5aR1 expression pattern with gestational age, birth weight, gender or infection.

In addition, among the mother-infant pairs, we did not see a correlation of receptor expression between the infants and their respective mothers.

#### 2.2.3. Higher C5aR2 Expression Neutrophils of Preterm and Term Infants as Compared to Healthy Adults

We also analyzed the intracellular expression pattern of C5aR2 and found an elevated expression in neutrophils from term and preterm born infants and mothers compared to adult controls (Figure 5).

We also found significant differences in the distribution of C5aR2 on certain NK cell subsets. CD56^bright^ CD16^+^ NK cells of preterm and term born infants show a lower individual expression of C5aR2 intracellularly, as measured by MFI, compared to mothers and adults (Kruskal–Wallis test *p* =< 0.0001, Figure 6a). On CD56^bright^ CD16^-^ NK cells, the MFI of intracellular C5aR2 was significantly higher in preterm born infants compared to mothers, term infants and adults (Kruskal–Wallis test *p* =< 0.0001, Figure 6b).

## 3. Discussion

In this explorative study, preterm infants had a higher intracellular expression of C5aR1 on neutrophils compared to term infants and adults. This observed phenomenon may reflect developmental immaturity and therefore contribute to the risk of “immunoparalysis”, uncontrolled hyper-inflammatory states and adverse long-term outcomes (4). On the other hand, the delicate balance of immunological tolerance between mother and fetus is abrogated in the context of preterm delivery, and enhanced C5aR1 expression may be a result of processes leading to preterm birth.

In line with this, previous studies have shown that complement needs to be tightly regulated at the maternal–fetal interface for a favorable pregnancy outcome [32]. Increased levels of C5aR1 may be a result of the disruption of the complement balance due to or resulting in preterm birth.

In the context of infection, an uncontrolled activation of the complement system results in a massive generation of the anaphylatoxin C5a [19,23,33], which, paradoxically, leads to a deficiency in the bacteriocidic function of neutrophils [34,35,36]. In the early phase of infection, the neutrophils are hyperresponsive and overactive [34], leading to a cytokine storm followed by tissue injury and organ damage [22]. The organ damage has been shown to be mediated through C5aR1, which is upregulated on the cell surface of, e.g., kidneys and lungs by IL-6. In our human setting, we found no expression differences of C5aR1 in neutrophils of preterm infants born from an inflammatory context (amniotic infection syndrome with or without early-onset sepsis) as compared to infants without evidence for inflammation at birth.

After binding of C5a, C5aR1 associates with β-Arestin [37] and the complex is then internalized and induces intracellular signal transduction cascades (including ERK1/2), which in neutrophils lead to chemotaxis, degranulation and production of superoxide [38,39,40]. Massive C5a generation, acting on neutrophils for a longer period, however, causes a paralysis of these cells [34,41]. They appear hyporesponsive to activation and show deficiencies in many functions such as chemotaxis, respiratory burst and phagocytosis, as well as in binding of C5a due to a downregulation of C5aR1 on the surface [20,34,42,43]. The reduced expression of C5aR1 on blood neutrophils correlates inversely with the severity of sepsis, mortality rate and outcome [39,44]. The global dysfunction of neutrophils results in the impaired killing of bacteria and therefore in a loss of the innate immune response and “immunoparalysis” [41]. In the context of neonatal immunity, in particular preterm birth, functional data on complement activation leading to “immunoparalysis” is lacking [4]. Our descriptive data support the idea that the production of large amounts of C5a may cause an internalization of C5aR1, which in turn can be detected intracellularly in greater quantity. Vice versa, we detected a lower percentage of C5aR1 on the surface of preterm neutrophils compared to healthy adult donors.

Whether overactivated neutrophils correlate with organ damage in preterm infants is yet unknown; for example, Pataky et al. detected significantly elevated C5a levels in the cerebrospinal fluid of preterm infants compared to term neonates, without a higher rate of acute invasive infections in the preterm cohort [45]. Preterm infants are often faced with an imbalance between tolerance and defense mechanisms [46] and a stronger systemic inflammatory response (SIRS) than the compensatory anti-inflammatory response (CARS).

This imbalance may be sustained through C5a-dependent pathways [47].

Whether our observations translate into a decreased functionality of preterm neutrophils, as has been reported by different research groups [9,48], is yet unknown and requires further study. Increased C5a production may occur in the sustained inflammation of the preterm [4], similar to the elevated basal signaling tone of key mediators of inflammation such as MAP-kinase and NFκB [49], and may also be a feature of a more proinflammatory signature of preterm blood [6].

Higher levels of intracellular C5aR1 in preterm infants may be explained by a pending transfer to the cell surface due to immaturity. Therefore, C5aR1 might still be located intracellularly, where in neutrophils it is most likely not activated and therefore not functional. Additionally, lytic capacity and levels of circulating complement components are lower in preterm infants [31]. This may also be due to a complement consumption exceeding production [50].

The upregulated expression of C5aR2 in neutrophils from mothers, term and preterm born infants compared to adults may be due to the inflammatory processes caused by labor and delivery [32]. The consequence of these elevated C5aR2 levels, however, remains unclear as the function of the receptor is still elusive.

In addition to the expression in neutrophils, we found a higher extracellular and intracellular expression of C5aR1 on NK cells from preterm infants compared to term infants, their mothers and healthy adults.

Complement receptor expressions on NK cells have rarely been investigated, particularly in humans and with regard to the different NK cell subtypes [51,52]. Min et al. [53] detected *c5ar1*-mRNA and little protein expression of C5aR1 on CD56^bright^ and CD56^dim^ NK cells, which we were able to confirm in all our cohorts, with an increased expression in preterm infants.

As the different NK cell subtypes have different functions [54], we were interested in whether the upregulation in preterm infants could be related to a specific subset. Our data revealed that C5aR1 is only upregulated on the CD56^dim^ or negative subsets, which represent the majority of preterm NK cells [55,56].

The CD56^dim^ subset is more cytotoxic, has more perforin and granzyme, and a higher surface expression of CD16, which is responsible for cell activation and ADCC [57]. Additionally, the ability to produce proinflammatory chemokines as a response to target recognition is greater than in the CD56^bright^ subset [58]. Enhanced activation of these NK cells via C5aR1 in preterm infants could support the development of a proinflammatory response (SIRS), while, due to the generally diminished cytotoxic capacities of neonatal NK cells [59], pathogen elimination is less effective.

The CD56^-^CD16^+^ subset is barely existent in healthy adults, but more present in neonates and other patients with an impaired immune function, such as in HIV infection or cytokine therapy [17]. These NK cells were shown to have reduced perforin expression and less capacity to ADCC but were able to produce chemokines such as MIP-1ß [18].

The CD56^bright^ subset, on the other hand, is known to be more immunoregulatory, producing anti-inflammatory cytokines such as IL-10 and INF-γ [60]. In preterm infants, this subset is not only less frequent ([56] and our own data), but also shows a reduced expression of C5aR1 compared to adults, resulting in less activation and a reduced anti-inflammatory response [61].

Preterm infants with early inflammatory complications (EOS or AIS) display a downregulated expression of C5aR1 on the surface of their NK cells. A downregulation of C5aR1 on neutrophils during sepsis [20,62] inversely correlated with sepsis severity. An association between sepsis and C5aR1 expression has also been shown for NK cells [51]. Enhanced activation of C5aR1 during EOS may result in internalization and a reduced surface expression of the receptor. Findings of a hyporesponsive status of NK cells during sepsis [59] support this theory of C5aR1-mediated immunoparalysis. Additionally, elevated C5a concentrations have been detected in the amniotic fluid of women with microbial invasion of the amniotic cavity and preterm birth [63].

Rahman Qazi et al. [16] showed that extremely low birthweight infants, developing sepsis in the first 14 days of life, also display NK cells with a higher CD56^dim^:CD56^bright^ ratio, while Bochenek et al. [55] observed a significantly decreased amount of immunoregulatory CD56^bright^ NK cells in preterm infants with LOS. We showed that C5aR1 is elevated mainly on the CD56 dim and negative subsets.

Our data support the hypothesis of a higher basal level of inflammation in preterm infants possibly resulting in sustained inflammation (SI) causing long-term morbidities and higher susceptibility to hyperinflammatory reactions with an increased sepsis risk.

To our knowledge, this is the first study investigating gestational-age-dependent differences in the extra- and intracellular expression of C5aR1 and C5aR2 on peripheral blood leukocytes. The study is hypothesis-generating for further studies to explain the pro-inflammatory state that preterm infants are often confronted with [46,64,65]. The limitations of our study are the explorative, descriptive and single-center study design and relatively small cohort size. Further functional data are needed to evaluate a causal link between higher C5aR1 expression in preterm neutrophils or NK cells and impairment of cellular function. We only evaluated samples on day 1–3 of life, which may not predict expression patterns in periods of sepsis. No direct association between the inflammatory context of preterm birth or early-onset sepsis with C5aR1 patterns in neutrophils was found. For outcome-related differences in C5aR1 expression, i.e., late-onset sepsis patients vs. unaffected controls, larger cohort studies are required. We acknowledge that our phenotypic immune cell analysis is limited, as for example NK cells can express several other receptors that are clonally distributed (such as KIR, CLIR and ILT members). Future studies on detailed analyses using mass flow cytometry or chip cytometry can give some additional insights.

As the experimental blockade of C5a and C5aR1 has been shown to result in an improved sepsis outcome [30], therapeutic targets of the complement system in human sepsis therapy are the focus of current research [66] and seem to be a promising approach.

Further investigation is needed to better understand the role of C5a in the innate immunity of preterm infants and to evaluate C5aR1 as a potential diagnostic and therapeutic target for this cohort.

## 4. Materials and Methods

### 4.1. Study Cohort

We performed a convenience sample single-center study at the Department of Pediatrics of the University Hospital Schleswig Holstein (UKSH), Campus Lübeck, recruiting preterm infants born between July 2019 and May 2020 and their mothers as part of our IRoN (Immunoregulation of the Newborn) study project at the UKSH Campus Lübeck. Preterm infants of a gestational age <32 + 0 weeks were included in their first three days of life. Exclusion criteria were lethal malformations and missing written consent. We also included term born infants and healthy adult donors as control groups.

### 4.2. Ethics

Informed written consent was obtained from all parents or legal representatives on behalf of the enrolled infants as well as from the adult donors.

The study parts were approved by the local committee on research in human subjects at the University of Lübeck (IRON AZ 15-304).

The blood withdrawal was performed only within a medically required blood withdrawal. Blood samples were obtained according to current guidelines of the European Medical Agency on the investigation of medicinal products in term and preterm infants, Committee for Medicinal Products for Human Use and Pediatric Committee (PDCO, 2009)

### 4.3. Flow Cytometry

EDTA whole blood samples (100 μL) were stored at room temperature and processed within 24 h of withdrawal. To distinguish between intra- and extracellular expression of C5aR1, we split the blood sample into two portions of 50 µL each, and stained one for extracellular/surface expression, while the other was fixed, permeabilized and stained for intracellular expression.

Plasma from 50 µL whole blood was removed via centrifugation and blood cells were stained using fluorochrome-labeled antibodies to characterize the different cell populations and the expression of the C5a-receptors. We used cell permeabilization and fixation reagents (Cell Staining Buffer, Fixation Buffer and Intracellular Staining Permeabilization Wash Buffer (10X), BioLegend^®^, San Diego, CA, USA) and the corresponding protocols for intracellular and extracellular staining [67,68].

To ensure cell viability after 24 h, we used a Viability Staining in a dilution 1:1000 (Fixable Viability Dye eFlour^®^780 conjugated, eBioscience, Thermo Fisher Scientific, Waltham, MA, USA) showing over 95% viability.

Every sample was stained with surface antibodies specific for CD16 (Alexa Flour 700 anti-human CD16, BioLegend^®^, San Diego, CA, USA; 1:100), CD19 (PerCP anti-human CD19, BioLegend^®^, San Diego, CA, USA, 1:100), CD56 (PE/Cy7 anti-human CD56, BioLegend^®^, San Diego, CA, USA; 10:100), CD66b (FITC anti-human CD66b, BioLegend^®®^, San Diego, CA, USA; 1:100) and CD3 (Brilliant Violet 421 anti-human CD3, BioLegend^®®^, San Diego, CA, USA; 3:100).

For every sample, a 50 μL aliquot was stained each for extracellular and intracellular C5aRs (APC anti-human CD88 (C5aR), BioLegend^®®^, San Diego, CA, USA, 1:100, PE anti-human C5L2, BioLegend^®^, San Diego, CA, USA 1:100).

Flow Cytometric analysis was performed within 24 h with a BD LSR II cytometer. The data were analyzed using FACS Diva Software Version 8.01 (BD Bioscience, San Jose, CA, USA) and Flow Jo™ version 10.7.1 (BD, Ashland, OR, USA).

The gating strategy is shown in Appendix A. To calculate the compensation, we used compensation beads, and to determine the threshold values for gating and to identify the fluorescence spread, we used fluorescence minus one (FMO) controls.

### 4.4. Clinical Data

*Gestational age (GA)* was calculated via obstetric examination and early prenatal ultrasound from the attending obstetrician.

*Suspected sepsis* was diagnosed by the attending pediatrician, when the infant received five days of antibiotic treatment and showed at least two clinical signs of systemic inflammation such as body temperature >38 °C or <36.5 °C, tachycardia >200/min, increased frequency of bradycardia or apnea, hyperglycemia >140 mg/dL, base excess >−10 mmol/L, increasing oxygen need and one conspicuous laboratory parameter such as CRP >5 mg/L, number of thrombocytes <100/nL, immature/total neutrophil ratio >0.2, number of leucocytes <5/nL).

*Blood-culture-confirmed sepsis* was defined as sepsis with at least two clinical signs, one laboratory parameter and confirmation of causative agent in blood culture.

*Early-onset sepsis (EOS)* was defined as sepsis (suspected or blood culture confirmed) occurring in the first three days of life.

*Amniotic infection syndrome (AIS*) was defined as a perinatal maternal temperature over 38 °C with at least one additional elevated maternal inflammatory marker (CRP > 10 mg/L or a number of leucocytes >16,000/μL) without other explanation, premature rupture of membranes, fetal or maternal tachycardia or foul-smelling amniotic fluid.

### 4.5. Statistical Analysis

#### Data Analysis Was Performed Using GraphPad Prism^®®^

After testing for normal distribution, an ANOVA (Kruskal–Wallis test) followed by a multiple comparisons test was performed to analyze the differences between groups.

Data were expressed as scatter plots showing the median and interquartile ranges. For the analysis of correlations, a Spearman *p* test was used. A *p* level < 0.05 was considered significant (* *p* < 0.05, ** *p* < 0.01, *** *p* < 0.001, **** *p* < 0.0001).

## 5. Conclusions

Our study revealed an elevated expression of the complement receptor C5aR1 on preterm infants’ neutrophils and NK cells, specifically the CD56^dim^ or negative subsets, compared to their mothers, term infants or adults.

Further investigations are needed to better understand the functional relevance of these observations and to correlate C5aR1 levels with patient-relevant outcomes.

## Figures and Tables

**Figure 1 ijms-24-10321-f001:**
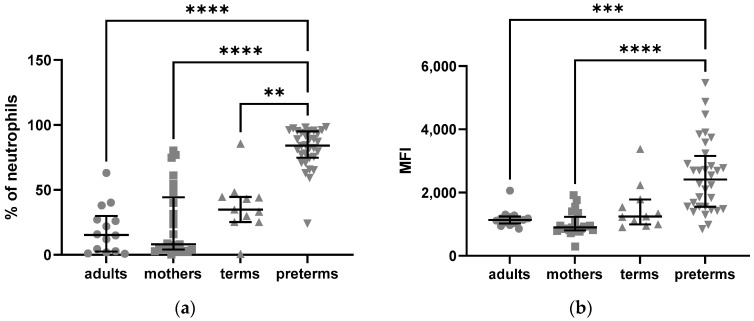
Intracellular expression of C5aR1 in neutrophils from adults, mothers, term and preterm infants assessed by flow cytometry: (**a**) frequencies of C5aR1-positive neutrophils; (**b**) mean fluorescence intensity (MFI) of intracellular C5aR1 in neutrophils. Kruskal–Wallis test, Tukey’s multiple comparisons, ** = *p* < 0.01, *** = *p* < 0.001, **** = *p* < 0.0001, bars indicate median and interquartile range.

**Figure 2 ijms-24-10321-f002:**
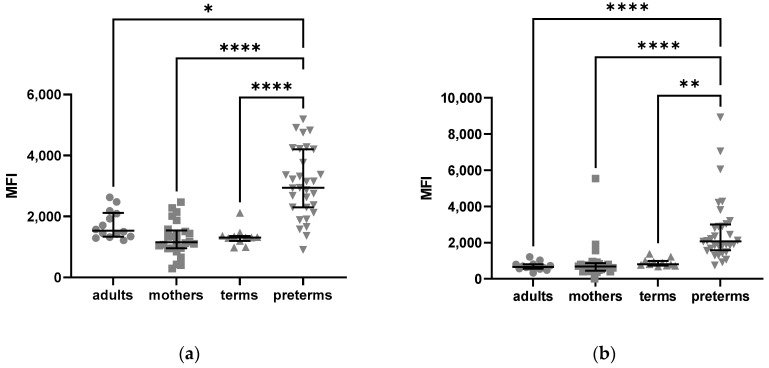
Expression of C5aR1 on NK cells from adults, mothers, term and preterm infants: (**a**) mean fluorescence intensity (MFI) of extracellular C5aR1 on NK cells; (**b**) MFI of intracellular C5aR1 on NK cells. Kruskal–Wallis test, Tukey´s multiple comparisons, * = *p* < 005, ** = *p* < 0.01, **** = *p* < 0.0001, bars indicate median and interquartile range.

**Figure 3 ijms-24-10321-f003:**
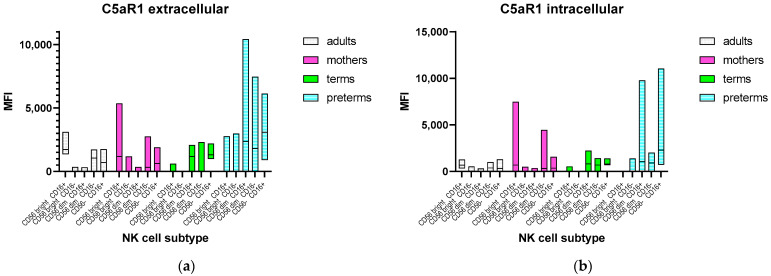
Expression of C5aR1 on the different subsets of NK cells from adults, mothers, term and preterm infant: (**a**) mean fluorescence intensity (MFI) of extracellular C5aR1 on NK cells; (**b**) MFI of intracellular C5aR1 on NK cells. Bars indicate median and minimum/maximum of C5aR1 expression to visualize differences between the subsets.

**Figure 4 ijms-24-10321-f004:**
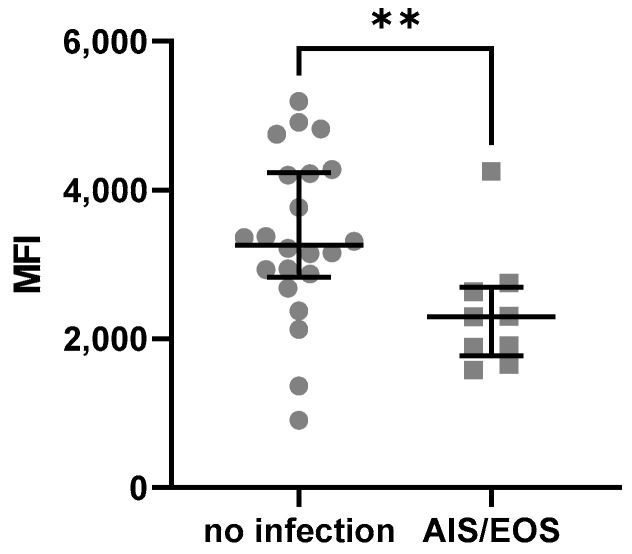
Extracellular expression of C5aR1 on NK cells from preterm infants with or without perinatal infection: mean fluorescence intensity (MFI) of extracellular C5aR1 on NK cells. Mann–Whitney test, ** = *p* < 0.01, bars indicate median and interquartile range.

**Figure 5 ijms-24-10321-f005:**
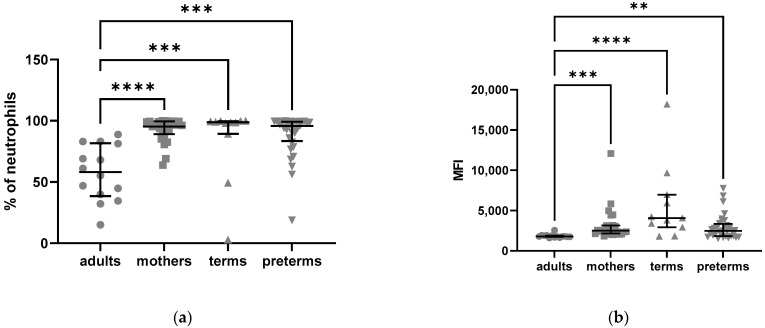
Intracellular expression of C5aR2 in neutrophils from adults, mothers, term and preterm infants assessed by flow cytometry: (**a**) frequencies of C5aR1-positive neutrophils; (**b**) mean fluorescence intensity (MFI) of intracellular C5aR1 in neutrophils. Kruskal–Wallis test, Tukey´s multiple comparisons, ** = *p* < 0.01, *** = *p* < 0.001, **** = *p* < 0.0001, bars indicate median and interquartile range.

**Figure 6 ijms-24-10321-f006:**
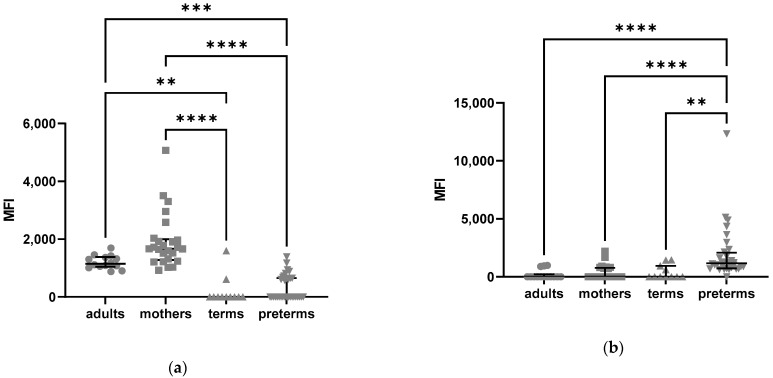
Intracellular Expression of C5aR2 in NK cells from adults, mothers, term and preterm infants assessed by flow cytometry: (**a**) mean fluorescence intensity (MFI) of C5aR2-positive CD56^bright^ CD16^+^ NK cells intracellularly; (**b**) mean fluorescence intensity (MFI) of C5aR2-positive CD56^bright^ CD16^-^ NK cells intracellularly. Kruskal–Wallis test, Tukey´s multiple comparisons, ** = *p* < 0.01, *** = *p* < 0.001, **** = *p* < 0.0001, bars indicate median and interquartile range.

**Table 1 ijms-24-10321-t001:** Characterization of cohorts.

Preterm infants (n)	32
Gestational age, weeks of gestation (mean/median/SD)	29.5/29.9/2.69
Birth weight, g (mean/median/SD)	1331.6/1300.0/464.13
Gender male (n/%)	14/43.8
Multiples (n/%)	10/31.3
AIS (n/%)	8/25.0
EOS (n/%)	1/3.1
**Mothers** (n)	25
Age, years (mean/median/SD)	29.0/29.0/6.09
Delivery at gestational age, weeks of gestation (mean/median/SD)	29.7/29.9/2.68
AIS yes (n/%)	6/24.0
**Term infants** (n)	11
Sex, male (n/%)	4/36.4
**Adults** (n)	14
Sex, male (n/%)	6/42.9
Age, years (mean/median/SD)	28.2/25.0/5.96

Legend: AIS—amniotic infection syndrome; EOS—early-onset sepsis.

## Data Availability

Data sharing will be made possible upon request.

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
