# Peer review of "Increased Expression of Anaphylatoxin C5a-Receptor-1 in Neutrophils and Natural Killer Cells of Preterm Infants"

_ijms, 2023, doi:10.3390/ijms241210321_

Round 1

Reviewer 1 Report (Previous Reviewer 3)

This is a very interesting article.

Reviewer 2 Report (Previous Reviewer 1)

The author have answered all the comments.

This manuscript is a resubmission of an earlier submission. The following is a list of the peer review reports and author responses from that submission.

Round 1

Reviewer 1 Report

The authors analyzed the expression pattern of C5a receptors on immune cells isolated from the blood of preterm infants compared with those of their mothers.

The paper is interesting as this topic is always up for discussion.

However, the paper needs improvement.

Title: "Elevated C5a-Receptor- expression drives a complement-mediated pro-inflammatory response in preterm infants".

The title is too exact, i.e., the authors state that C5a- receptor expression drives the complement-mediated pro-inflammatory response, although they did not perform any in vitro experiment; thus, this needs to be changed. Moreover, they hypothesize at the end or in the conclusion section of the article: "We hypothesize that the enhanced expression of C5aR1 in neutrophils and on specific subsets of NK cells shown in our study promotes pro-inflammatory chemokine production...".

Please, rephrase the title.

Abstract: the abstract needs to be more concise. It should include the background and objective, then the methods and results, and at the end the summary. The conclusion of your abstract again looks like an introduction or more like a discussion. The abstract needs to have its proper flow and be written very clearly, as it is the most important part of the article it presents.

Also, the number of participants should be mentioned.

Introduction: satisfactory.

Methods: inclusion and exclusion criteria should be mentioned (for infants and mothers), especially the presence of inflammatory markers

Conclusion: it looks more like a discussion and not a conclusion. Please be more clear here.

Author Response

Response to Reviewer 1 Comments

Point 1: The title is too exact, i.e., the authors state that C5a- receptor expression drives the complement-mediated pro-inflammatory response, although they did not perform any in vitro experiment; thus, this needs to be changed. Moreover, they hypothesize at the end or in the conclusion section of the article: "We hypothesize that the enhanced expression of C5aR1 in neutrophils and on specific subsets of NK cells shown in our study promotes pro-inflammatory chemokine production...".
Please, rephrase the title.

Response 1: We rephrased the title.

Point 2: Abstract: the abstract needs to be more concise. It should include the background and objective, then the methods and results, and at the end the summary. The conclusion of your abstract again looks like an introduction or more like a discussion. The abstract needs to have its proper flow and be written very clearly, as it is the most important part of the article it presents.

Response 2: We rewrote the abstract to more closely follow the proper set up and the conclusion to be less diffuse.

Point 3: Also, the number of participants should be mentioned.
Response 3: We added the number of participants in the abstract.

Point 4: Introduction: satisfactory.

Response 4: Thank you.

Point 5: Methods: inclusion and exclusion criteria should be mentioned (for infants and mothers), especially the presence of inflammatory markers

Response 5:  Exclusion criteria (lethal malformations and missing consent) were added to the text. Inflammatory marker were not relevant for inclusion in the IRoN study.

Point 6: Conclusion: it looks more like a discussion and not a conclusion. Please be more clear here.

Response 6: The conclusion has been rewritten to be more concise.

Reviewer 2 Report

Dear Author, 

Your article Elevated C5a-Receptor-expression drives a complement-medi- 2 ated pro-inflammatory response in preterm infants is intersting and Quality of Presentation. My desision is accept in present form! 

Author Response

Thank you very much!

Reviewer 3 Report

This is a paper about the complement-mediated pro-inflammatory response in preterm infants caused by elevated C5a-Receptor-expression. It is a well-written paper on a very interesting field.

COMMENTS:

1.       Line 331: “July” instead of “Juli”

2.       Line 378: “Suspected sepsis” instead of “sepsis” if no positive blood-cultures weren’t obtained.

3.       Line 387: Samples from 1-3 days of life were evaluated. Yet, a  LOS cohort appears in Table 1.

Author Response

Response to Reviewer 3 Comments

Point 1: Line 331: “July” instead of “Juli”

Response 1: Changed

Point 2: Line 378: “Suspected sepsis” instead of “sepsis” if no positive blood-cultures weren’t obtained.

Response 2: Changed  

Point 3:   Line 387: Samples from 1-3 days of life were evaluated. Yet, a  LOS cohort appears in Table 1.

Response 3: LOS war mentioned to characterize the cohort with regard to the infection vulnerability. But as it is of course not strictly relevant for this paper, we removed it from the table.

Round 2

Reviewer 1 Report

The authors implemented all necessary changes.